# Identification and Quantification of Adulterants in Coffee (*Coffea arabica* L.) Using FT-MIR Spectroscopy Coupled with Chemometrics

**DOI:** 10.3390/foods9070851

**Published:** 2020-06-30

**Authors:** Mauricio Flores-Valdez, Ofelia Gabriela Meza-Márquez, Guillermo Osorio-Revilla, Tzayhri Gallardo-Velázquez

**Affiliations:** 1Instituto Politécnico Nacional, Escuela Nacional de Ciencias Biológicas-Santo Tomás, Departamento de Biofísica, Prolongación de Carpio y Plan de Ayala S/N, Col. Santo Tomás, Alcaldía Miguel Hidalgo, Ciudad de México C.P. 11340, Mexico; mfv@live.com; 2Instituto Politécnico Nacional, Escuela Nacional de Ciencias Biológicas-Zacatenco, Departamento de Ingeniería Bioquímica, Av. Wilfrido Massieu S/N, Esq. Cda, Miguel Stampa, Col. Unidad Profesional Adolfo López Mateos, Zacatenco, Alcaldía Gustavo A. Madero, Ciudad de México C.P. 07738, Mexico; ogmmz@yahoo.com.mx (O.G.M.-M.); osorgi@gmail.com (G.O.-R.)

**Keywords:** *Coffea arabica* L., FT-MIR, chemometric studies, coffee quality

## Abstract

Food adulteration is an illegal practice performed to elicit economic benefits. In the context of roasted and ground coffee, legumes, cereals, nuts and other vegetables are often used to augment the production volume; however, these adulterants lack the most important coffee compound, caffeine, which has health benefits. In this study, the mid-infrared Fourier transform spectroscopy (FT-MIR) technique coupled with chemometrics was used to identify and quantify adulterants in coffee (*Coffea arabica* L.). Coffee samples were adulterated with corn, barley, soy, oat, rice and coffee husks, in proportions ranging from 1–30%. A discrimination model was developed using the soft independent modeling of class analogy (SIMCA) framework, and quantitative models were developed using such algorithms as the partial least squares algorithms with one variable (PLS1) and multiple variables (PLS2) and principal component regression (PCR). The SIMCA model exhibited an accuracy of 100% and could discriminate among all the classes. The quantitative model with the highest performance corresponded to the PLS1 algorithm. The model exhibited an R^2^c: ≥ 0.99, standard error of calibration (SEC) of 0.39–0.82, and standard error of prediction (SEP) of 0.45–0.94. The developed models could identify and quantify the coffee adulterants, and it was considered that the proposed methodology can be applied to identify and quantify the adulterants used in the coffee industry.

## 1. Introduction

Coffee (*Coffea arabica* L.) is one of the most widely consumed beverages worldwide, second only to water and tea, and it is classified as the second most commercialized product worldwide after petroleum [1]. Most coffee is produced in Brazil, Vietnam and Colombia. Mexico is also an important coffee producer and consumer, with a per capita consumption of 1 to 2.9 kg and an export-based annual income of 900 million dollars [2].

Coffee beans are used to prepare the beverage, coffee, which is consumed worldwide for several reasons (pleasure, health benefits and improvement in physical performance) [3]. Due to the commercial importance of coffee production and consumption, the product is frequently adulterated [4]. Coffee adulteration has been a widespread practice performed primarily to elicit economic benefits. Specifically, coffee adulteration is performed to counter high prices or product scarcity or to lower the production costs [5]. Twigs, coffee husks, roots, legumes and other roasted grains such as corn, barley, soy, oat and rice have been used as coffee adulterants because they are relatively inexpensive, and their chemical composition and organoleptic properties do not drastically affect those of coffee. Over 100 products have been estimated to be used as coffee adulterants, and although this practice is accepted by consumers in certain countries, such as England, selling coffee mixed with substances such as chicory (French coffee) or figs (Viennese coffee) is illegal if this mixing is not declared by the producer [6].

In recent years, several methods have been developed to analyze the authenticity of coffee. With the considerable increase in the coffee industry, it is necessary to develop techniques to identify the adulterants in the product [4]. Several analytical techniques to identify and quantify adulterants in coffee have been developed, for example, anion exchange chromatography [7], capillary electrophoresis [8], near infrared Fourier transform spectroscopy (FT-NIR) [9], high-performance liquid chromatography (HPLC) [10] and UV-vis spectroscopy [11]. Although these techniques are sensitive and precise, they can be performed only by trained personnel, and the sample preparation time may be extremely large. In contrast, mid-infrared Fourier transform spectroscopy (FT-MIR) is a rapid technique that does not require sample preparation, and it is considered ecofriendly because it does not require reagents or solvents [12]. Although FT-NIR is also a fast approach, it involves overtone information and a combination of fundamental vibrations, which can be difficult to interpret; consequently, the findings may be less reproducible and specific. Moreover, the bands in the near infrared region are usually weak in intensity and exhibit an overlap, which renders them less useful than the mid-infrared region for analysis. In addition, compared to the near-infrared technique, the mid-infrared region is highly sensitive to the precise chemical composition of the samples [13].

The FT-MIR spectroscopy technique coupled with chemometrics has been successfully used to perform the quality control analysis and detection of adulterants in several food products [14]. Nevertheless, the quantification of adulterants in coffee by using a combined FT-MIR spectroscopy and chemometrics approach has not been reported, although this technique has been used to quantify the amount of caffeine in coffee solutions [15] and discriminate between defective and intact roasted coffee beans [16].

Considering this background, the objective of this study was to develop chemometric models based on FT-MIR spectroscopy to identify and quantify the amount of coffee husks, corn, barley, soy, oat and rice as adulterants in roasted and ground coffee (*Coffea arabica* L.).

## 2. Materials and Methods 

### 2.1. Samples

Parchment coffee (*Coffea arabica*) was acquired from Huatusco, Veracruz, México (19°09′ N 96°58′ W). Coffee husks (*Coffea arabica*), barley (*Hordeum vulgare*), corn (*Zea mays*), soy (*Glycine max*), oat (*Avena sativa*) and rice (*Oryza sativa*) were acquired from grocery stores in Mexico City. The quality of all the samples was assured by the supplier.

### 2.2. Adulterated Samples Preparation 

Coffee beans, coffee husks, barley, corn, soy, oat and rice samples (30 g) were roasted in a coffee bean roasting device (NESCO^®^ model CR-1000)—roasting cycle: 15 min, cooling cycle: 5 min, maximum temperature: 250 °C). After roasting, the samples were ground (2 cycles of 30 s) and sifted (300 μm). These conditions were selected because the final samples have been noted to resemble the adulterated coffee available in the market [7,17,18]. Subsequently, coffee was adulterated using coffee husks, corn, barley, soy, oat and rice in a binary mixture. 

A total of 180 samples were adulterated (thirty coffee samples with each adulterant) in concentrations ranging from 1 to 30% (w/w) in increments of 1%. The minimum percentage of adulteration was considered to be 1%, because certain international regulations allow the presence of foreign bodies in coffee [19]. The 30% adulteration limit was set by García et al. [7], Oliveira et al. [17] and Reis et al. [18], who reported that the proportion of coffee adulterants is usually less than 30%. Furthermore, thirty unadulterated coffee samples were roasted, grinded and sifted, following the aforementioned procedure.

### 2.3. Acquisition of FT-MIR Spectra

The infrared spectra of unadulterated coffee, adulterants and adulterated samples were obtained using an infrared Fourier transformed spectrophotometer (FT-MIR) (Spectrum GX, Perkin Elmer, Waltham, MA, USA) equipped with a deuterated triglycine sulfate detector and diamond crystal sampling accessory. The FT-MIR spectra were obtained in a range of 4000–650 cm^−1^ with 64 scans at a resolution of 2 cm^−1^ and in units of absorbance (A). Moreover, the spectra were collected using an air spectrum as the background. Approximately 3 mg of each sample was collected in the diamond crystal sampling accessory and registered in triplicate using the Spectrum software version 3.01.00 (PerkinElmer^®^, Waltham, MA, USA).

### 2.4. Multivariate Analysis 

#### 2.4.1. Discrimination Model 

A discrimination model (soft independent modeling of class analogy, SIMCA) was developed to distinguish between the adulterated and unadulterated samples. The principal component analysis (PCA) was performed to realize class discrimination. Seven classes were created—(pure coffee, coffee–coffee husks, coffee–corn, coffee–barley, coffee–soy, coffee–oat, coffee–rice). A total of 140 and 70 FT-MIR spectra (20 and 10 spectra from each class, respectively) were utilized for the model calibration and validation, respectively. The FT-MIR spectra were registered into the software AssureID version 3.0.0.0132 (PerkinElmer^®^, MA, USA) to discriminate between the adulterated and pure samples. The model was optimized by performing the following spectral pretreatments—blank samples (4000–3500, 2800–1800, and 800–650 cm^−1^), filters (to eliminate CO_2_ and H_2_O), normalization (multiplicative scatter correction, MSC), Savitzky–Golay filter (9 and 19 points for smoothing) and baseline correction (offset).

The predictive ability of the SIMCA model was evaluated considering the projection of the first three principal components (PCs), which correspond to the class separation (the interclass distance, which indicates the similarity between classes, must be higher than 3), recognition percentage (sensitivity) and rejection percentage (selectivity). If the rejection and recognition percentage are 100%, it is considered that the SIMCA model discriminates the classes adequately. The model was validated considering the total distance (a value less than 1 indicates that the sample has been properly classified) and residual distance (a value higher than 3 indicates the sample has a variation source that has not been found) [20].

#### 2.4.2. Quantitative Model 

To quantify the adulterant percentage, a matrix was created to mathematically relate the FT-MIR spectra with the adulteration percentage. Six chemometric models were developed (coffee–coffee husks, coffee–corn, coffee–barley, coffee–soy, coffee–oat, coffee–rice). Thirty FT-MIR spectra of each adulteration system along with the adulteration percentages (1–30%) were put into the multidimensional statistical analysis program Spectrum QUANT+ version 4.51.02 (PerkinElmer, Inc.). The QUANT+ software employed three algorithms—partial least squares with one (PLS1) and multiple variable (PLS2) and principal component regression (PCR). 

The model optimization involved performing the following spectral pretreatments—normalization (multiplicative scatter correction, MSC), Savitzky–Golay filter (5 points for smoothing) and baseline correction (offset). The model with the highest predictive ability was selected based on the latent variables (factors), coefficient of determination of calibration (R^2^c, must be close to 1) and standard error of calibration (SEC, must be as low as possible) [21]. The model was validated using with 10 FT-MIR spectra of each adulteration system, and the following statistic parameters were evaluated—coefficient of determination of validation (R^2^v, must be as close to 1 as possible), standard error of prediction (SEP, must be as low as possible), Mahalanobis distance (indicates the spectral similarities between samples and must be lower than 1) and residual ratio (must be lower than 3; a higher value suggests that the characteristics of the sample are different from those of the samples used in the set calibration) [21].

## 3. Results and Discussion

### 3.1. Interpretation of Spectra FT-MIR

Figure 1 shows an example of the FT-MIR spectra of pure coffee adulterated with each adulterant with a concentration of 30%. The first band located at 3470 cm^−1^ is attributed to the O-H stretching, which is associated with the presence of water in the matrix or carboxyl groups present in compounds such as chlorogenic acid, which is present in coffee. The following band at 3008 cm^−1^ can be attributed to the stretching of the C=C cis double bond present in lipids. The peaks at 2920–2855 cm^−1^ correspond to the symmetric and asymmetric stretching of a C-H bond present in the methyl and methylene groups in polysaccharides, which belong to lignin, a characteristic polymer in coffee beans [22].

A peak also occurs at 1746 cm^−1^ owing to the stretching of the ester group OC=O of quinic acids (C_7_H_12_O_6_) formed during coffee roasting [23]. The peak at 1704 cm^−1^ represents the vibrations of the carbonyl group C=O present in free fatty acids [24]. The band at 1658 cm^−1^ can be attributed to the stretching of the C=O group present in amides. The band at 1608 cm^−1^ is associated with vibration of the C-N group [25], which is likely related to compounds such as caffeine, trigonelline or nicotinic acid. The bands at 1460–1377 cm^−1^ originate from the bending vibrations of the C-H bond present in aliphatic groups [24]. The peak at 1237 cm^−1^ can be attributed to the carboxyl group in the chlorogenic acid. Two bands occur between 1155 and 1028 cm^−1^, which are related to the vibrations of the C-O bonds in the ester groups found in chlorogenic acid or polysaccharides such as lignin, pectin or starch [26]. Finally, the last band at 869 cm^−1^ is related to the bending vibrations of the C-H bonds in alkenes [22], likely related to the unsaturated fatty acids or unsaturations present in coffee. 

The FT-MIR spectra of pure coffee and adulterated samples exhibited a spectral variability, which is fundamental to build chemometric models.

### 3.2. Multivariable Analysis 

#### 3.2.1. Discrimination Models

The SIMCA model was developed considering the spectral regions between 3500–2800 cm^−1^ and 1800–800 cm^−1^ because these regions corresponded to spectral variability, which is necessary to discriminate among classes. The SIMCA model yielded an appropriate spatial distribution of the classes in terms of the first three principal components (Figure 2), thereby indicating that no atypical or overlapped samples were detected. The samples were considered to be properly assigned to the class that they belonged to. The elliptic spaces (clouds) represented the 99% confidence interval and indicated that the samples were a part of the class [20].

This spatial distribution was reflected through the interclass distance (Table 1). The value of this parameters was higher than 3, indicating that the classes could be discriminated. In general, an interclass distance of 0 means that the groups are identical, and a distance larger than 3 indicates that the samples are separated and thus different [20]. These results demonstrate the capability of the developed SIMCA model to discriminate the spectral signal of pure coffee with respect to that of adulterated coffee, based on the FT-MIR spectral differences between the clusters.

Table 2 presents the results of the classification performance of the SIMCA model based on the percentage recognition (sensitivity) and rejection (selectivity). A 100% rate of recognition was obtained for each group, thereby indicating that all twenty samples of each group were properly classified and, similarly, all the 120 samples that did not belong to the group were rejected at a rate of 100%. These results demonstrate the capability of the SIMCA model to correctly classify all the examined groups.

The validation of the SIMCA model (Table 3) indicated that the employed samples were properly identified and classified, and the findings were in agreement with those of the statistical parameters. In general, a total distance less and more than 1 indicates that the sample has been properly classified or not classified, respectively [20]. Moreover, a residual distance more than 3 indicates that the sample contains a source of variation that has not been previously encountered [20]. The obtained results indicate that the SIMCA model possesses a high discrimination capacity.

The SIMCA model has been successfully applied previously, for instance, to discriminate between pure mezcal and adulterants [27], cow’s milk and tetracycline residues [28], and avocado oil and adulterants [29]. In all such cases, the SIMCA model has been proved to useful in classifying samples.

#### 3.2.2. Quantitative Models

Predictive models were created considering the spectral regions at 3500–2800 and 1800–800 cm^−1^ because these regions exhibited the highest spectral variability, corresponding to the change in absorbance and increase in the amount of adulterants in the samples. Table 4 presents the statistical parameters for the six developed predictive models.

The algorithm that led to the optimal predictive model was PLS1. In particular, the model developed using PLS1 exhibited the highest R^2^c value (0.99), which indicates the provision of excellent quantitative information [20]. Additionally, low SEC (0.39–0.82) values indicate a low error in the regression, since they have the same unit as the actual value. 

Table 4 also lists the number of factors considered. The number of factors used in the PLS or PCR models is a critical parameter. In general, the use of extremely few factors may generate an underfitted model that loosely fits the data; in contrast, using excessively many factors may generate an overfitted model involving excessive information (noise), thereby generating a low SEC albeit an inferior performance in the validation set. Bureau et al. [30] indicated that the maximum number of factors must be up to a maximum of 15. The models developed using PLS1 satisfied this requirement, and the number of factors was between 9 and 11, indicating that the developed models yielded minimal SEP values, thereby generating acceptable prediction results.

Figure 3 shows the scatter plots of the correlation between the actual and predicted values of the calibration set. All the points of the plots can be noted to fall on or close to the unity line (R^2^c: 0.99), indicating the provision of excellent quantitative information [21]; therefore, the models can be expected to yield excellent prediction results.

The robustness of the PLS1 models was investigated considering the prediction of 10 FT-MIR spectra of each adulteration system. Table 4 lists the Rv^2^ and SEP values between the predicted and actual values of the external samples; these parameters were used to evaluate the performance of the PLS1 models. The Rv^2^ between the predicted and actual values of the external samples was 0.99, indicating good prediction [21]. Moreover, the obtained SEP values ranged from 0.45–0.94 (Table 4), indicating an excellent prediction [21]. In general, the SEP is indicative of a model’s ability to accurately predict unknown samples. A high SEP indicates that the model cannot effectively predict the samples accurately. Therefore, models should exhibit low SEC and SEP values with only a small difference between the two values and a high R^2^. Large differences between the SEC and SEP values indicate the introduction of too many factors (latent variables) in the models, indicating the presence of noise [30].

Similarly, the Mahalanobis distance for each validation sample was less than 1.0 (Table 4), indicating that each sample was correctly classified by the models. It was thus considered that the calibration and external validation samples were reasonably similar. In general, the Mahalanobis distance can help determine the similarity of a set of values from an unknown sample to a set of values measured from a collection of known samples. This distance is measured in terms of the standard deviation from the mean of the training samples, and the reported matching values provide a statistical measure of how well the spectrum of the unknown sample matches (or does not match) the original training spectra [21]. Finally, the residual error was less than 3.0 (Table 4), indicating that the residual spectrum from the external validation samples contained features that were suitably described and modeled during the calibration [21].

Figure 4a–f shows the correlation between the actual values and the values predicted using the models developed using the PLS1 algorithm. It can be noted that regardless of the adulterant concentrations, which range between 1% and 30%, the predicted values fall very close to the equal concentration line, demonstrating the excellent prediction capability of the models.

The results indicated similarities between the calibration and validation samples; therefore, it was considered that the samples used to validate the PLS1 models exhibited the required spectral characteristics, which were modeled during the calibration. Consequently, the models could correctly predict the percentage of adulterants (coffee husks, corn, barley, soy, oat, rice) in samples that were not used to develop the calibration models.

## 4. Conclusions

The FT-MIR spectroscopy approach coupled with chemometrics could be used to identify pure and adulterated coffee samples with a 100% accuracy. The prediction models could quantify adulterants in coffee, with concentrations ranging from 1 to 30%. The developed models can be used in the coffee industry as a quality control tool to identify and quantify possible adulterants in coffee, thereby verifying the authenticity of the product. 

The development of other predictive models taking into account other possible adulterants is recommended to build a robust model that can predict several types of adulterants. Moreover, future work may focus on attaining lower detection limits for the adulterant of interest. It is also advisable to build predictive models pertaining to ternary or quaternary mixtures because coffee may be adulterated with more than one adulterant. 

## Figures and Tables

**Figure 1 foods-09-00851-f001:**
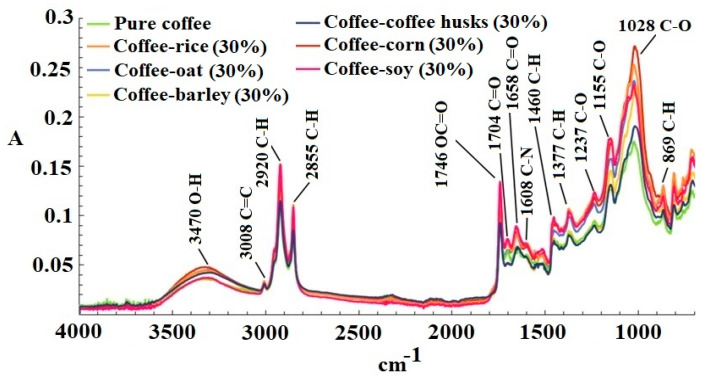
Infrared Fourier transformed spectrophotometer (FT-MIR) spectra of pure coffee adulterated with each adulterant with a concentration of 30%.

**Figure 2 foods-09-00851-f002:**
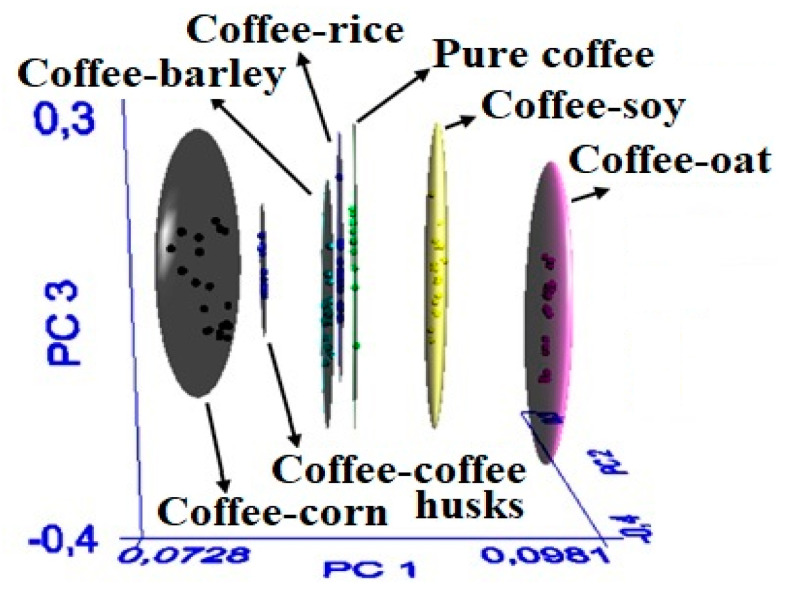
Spatial distribution of the classification SIMCA model.

**Figure 3 foods-09-00851-f003:**
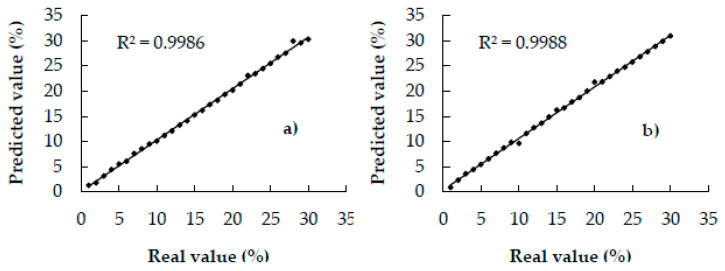
Scatter plots of actual and predicted values of coffee adulterated with—(**a**) coffee husks, (**b**) corn, (**c**) barley, (**d**) soy, (**e**) oat, and, (**f**) rice of calibration samples.

**Figure 4 foods-09-00851-f004:**
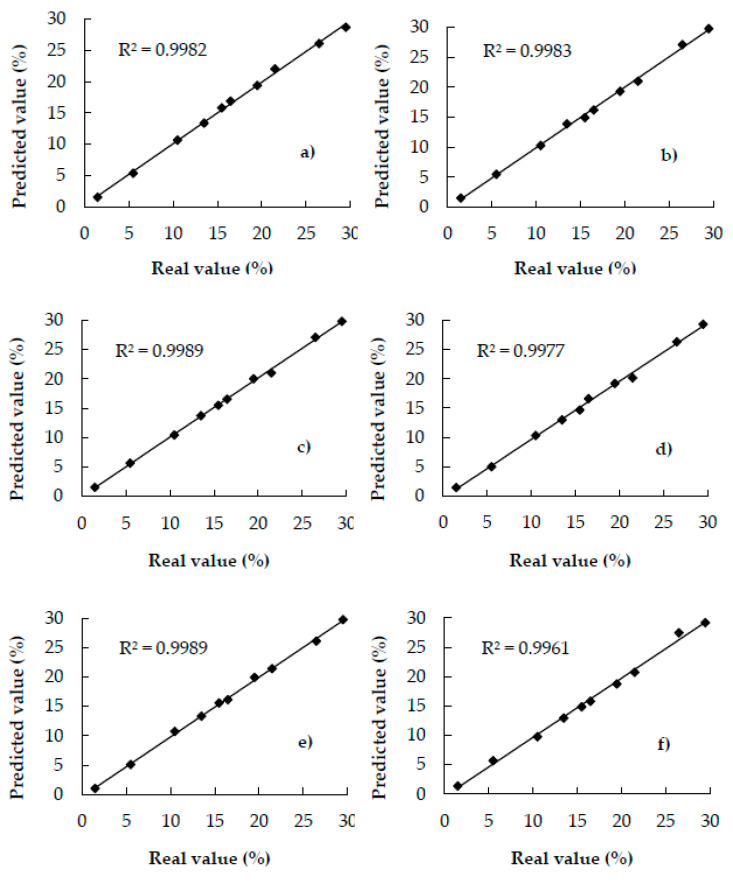
Scatter plots of actual and predicted values of coffee adulterated with—(**a**) coffee husks, (**b**) corn, (**c**) barley, (**d**) soy, (**e**) oat, and, (**f**) rice of validation samples.

**Table 1 foods-09-00851-t001:** Interclass distance of the soft independent modeling of class analogy (SIMCA) model.

	Coffee-Coffee Husks	Coffee-Corn	Coffee-Barley	Coffee-Soy	Coffee-Oat	Coffee-Rice
Pure coffee	3.04	7.67	16.10	8.61	13.3	5.09
Coffee-coffee husks	-	5.90	10.10	6.07	10.5	3.31
Coffee-corn	-	-	11.30	5.89	10.7	6.16
Coffee-barley	-	-	-	3.42	4.92	5.99
Coffee-soy	-	-	-	-	3.15	4.60
Coffee-oat	-	-	-	-	-	6.23

Interclass distance should be as high as possible, minimum 3.

**Table 2 foods-09-00851-t002:** Recognition (sensitivity) and rejection (selectivity) of the SIMCA model.

Class	Recognition (%)	Rejection (%)
Pure coffee	100 (20/20)	100 (120/120)
Coffee-coffee husks	100 (20/20)	100 (120/120)
Coffee-corn	100 (20/20)	100 (120/120)
Coffee-barley	100 (20/20)	100 (120/120)
Coffee-soy	100 (20/20)	100 (120/120)
Coffee-oat	100 (20/20)	100 (120/120)
Coffee-rice	100 (20/20)	100 (120/120)

Recognition and rejection must be 100%.

**Table 3 foods-09-00851-t003:** Validation results of the SIMCA model.

Specified Material ^a^	Identified Material ^b^	Result ^c^	Total Distance ^d^	Residual Distance ^e^
Pure coffee	Pure coffee	Identified	0.44–0.67	0.61–0.93
Coffee-coffee husks	Coffee-coffee husks	Identified	0.52–0.76	0.72–0.96
Coffee-corn	Coffee-corn	Identified	0.58–0.81	0.80–1.15
Coffee-barley	Coffee-barley	Identified	0.72–0.92	0.98–1.26
Coffee-soy	Coffee-soy	Identified	0.75–0.93	1.03–1.24
Coffee-oat	Coffee-oat	Identified	0.34–0.91	0.67–1.25
Coffee-rice	Coffee-rice	Identified	0.33–0.62	0.52–0.73

^a^ Specified material of validation; ^b^ identified material by SIMCA model; ^c^ result indicates if the sample was identified or rejected (does not belong to the model); ^d^ total distance indicates if the sample was classified correctly (must be less than 1); ^e^ high residual distance indicates that the sample contains a source of variation that has not been previously found (must be less than 3).

**Table 4 foods-09-00851-t004:** Calibration data from chemometric models to predict adulterants of coffee (*Coffea arabica* L.).

Calibration Set	Calibration (*n* = 30)				Validation (*n* = 10)			
	Algorithm	Factors ^a^	*R*^2^c ^b^	SEC ^c^	R^2^v ^d^	SEP ^e^	Mahalanobis Distance ^f^	Residual Ratio ^g^
Coffee-coffee husks	***PLS1***	***11***	***0.99***	***0.48***	0.99	0.57	0.33–0.82	1.28–1.17
	PLS2	14	0.96	2.25	-		-	-
	PCR	8	0.83	4.22	-		-	-
Coffee-corn	***PLS1***	***8***	***0.99***	***0.45***	0.99	0.51	0.49–0.81	1.11–1.87
	PLS2	13	0.97	1.73	-		-	-
	PCR	8	0.97	1.73	-		-	-
Coffee-barley	***PLS1***	***11***	***0.99***	***0.41***	0.99	0.60	0.45–0.49	0.96–1.36
	PLS2	14	0.83	4.77	-		-	-
	PCR	6	0.72	4.95	-		-	-
Coffee-soy	***PLS1***	***10***	***0.99***	***0.82***	0.99	0.94	0.21–0.56	0.96–1.48
	PLS2	8	0.87	3.65	-		-	-
	PCR	9	0.88	3.60	-		-	-
Coffee-oat	***PLS1***	***11***	***0.99***	***0.56***	0.99	0.91	0.46–0.62	1.18–2.07
	PLS2	8	0.69	5.50	-		-	-
	PCR	10	0.75	5.21	-		-	-
Coffee-rice	***PLS1***	***9***	***0.99***	***0.39***	0.99	0.45	0.37–0.72	1.67–1.85
	PLS2	10	0.97	1.83	-		-	-
	PCR	8	0.93	1.78	-		-	-

The algorithm with the best predictive results is highlighted in italics and bold. ^a^ Factors, ^b^ R^2^c (coefficient of determination of calibration, must be close to 1), ^c^ SEC (standard error of calibration, should be as low as possible), ^d^ R^2^v (coefficient of determination of validation, must be close to 1); ^e^ SEP (standard error of prediction, should be as low as possible); ^f^ Mahalanobis distance (must be less than 1), ^g^ Residual ratio (must be less than 3).

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
