# Peer review of "Identification and Quantification of Adulterants in Coffee (Coffea arabica L.) Using FT-MIR Spectroscopy Coupled with Chemometrics"

_foods, 2020, doi:10.3390/foods9070851_

Round 1

Reviewer 1 Report

This manuscript deals with an important and still current topic of adulterants in coffee. The experiment procedure was properly designed and according to proposed methodology to combine FT-MIR and chemometrics, obtained results was adequate to describe conclusions. The manuscript is well written and only a few suggestion need to be correct or clear.

Title clearly inform about the topic of manuscript. However, to differ keywords from the title, I suggest to use term 'chemometry studies', and add 'coffee quality.

Authors said that from 1 to 30% is the adulteration limit, but do we knoe acceptable trace contaminants for example for coffee husks, are there any norms, regulations? 

In subchapter 'Adulterated samples preparation' authors decribed that 180 samples were prepared, however in Figure 1 it is only one sample per each adulteration presented. It is an average out of 180 samples? FT-MIR look the same for every 1- 30% adulteration?

Reviewer 2 Report

Please improve the narrative in English. In many places it sounds very awkward and/or there are typos and mistakes. For example

  1. line 43 "production costs"
  2. Line 107 PC instead of CP
  3. lime 102 "consisted of"

However, the main reasons for my Reconsider After Major Revision decision are the following:

  1. Data that demonstrates that 1-30% adulteration of coffee with known adulterants can be quantitated is missing. Part 3.2.2 Quantitation Models give some predictive ability information however doesn't show results of quantitative prediction of actual samples with known adulteration levels. In the  Conclusions in line 227 it is stated that the prediction models can be applied to quantify adulterants. No such data was presented. I recommend adding this data and re-writing 3.2.2. 
  2. Data demonstrating accuracy of the method is missing. In the abstract it's stated that the accuracy is 100%, while inn he conclusions it's 99%
  3. Why only 1-30% adulterant range was selected. Does the model work beyond this range ?
  4. You mentioned specificity and sensitivity. Did you estimate them?

Reviewer 3 Report

The manuscript reports a method development and validation for Identification and quantification of adulterants in coffee by FT-MIR spectroscopy. The work is reasonable to perform as coffee adulterations have been frequently practiced for economic motivation. The title is suitable for the content of the work. For the keywords, it would be better to provide other keywords different than those mentioned in the title to improve the searchability of this work

In the introduction, FT-MIR is preferred over other techniques due to a fast technique. A further justification should be made as the aforementioned FT-NIR is also considered as a fast analytical technique. Additionally, the background of this study will be completed if the author could mention the reason why coffee husks, corn, barley, soy, oat, and rice are chosen to be studied as adulterants.

Material and Method
In the adulterated sample preparation, there is a lack of explanation on which stage the adulterants are added into the coffee samples and how to assure that the final samples can be mimic the adulterated coffee as in the market.

Part 2.3. The acquisition of FT-MIR spectra was performed by collecting the absorbance signal, why did this signal be used instead of reflectance for the solid sample?

Result and Discussion.
It was mentioned in M&M that "Infrared spectra of unadulterated coffee, adulterants, and adulterated samples were obtained using an Infrared Fourier Transformed Spectrophotometer" nonetheless in the section of 3.1. Interpretation of spectra FT-MIR, there are only results for unadulterated and adulterated coffee samples without adulterants solo.

Line 174, the authors confused about using the validation term of sensitivity, selectivity, and specificity. Recognition is not mostly related to sensitivity but selectivity.

The validation of the SIMCA model was evaluated on the basis of total distance and residual distance. The statistical calculations have been performed, however, no reference justifying that the total distance is less than 1 and residual distance is less than 3 for the acceptable values.

In general, the authors have not provided an explanation of what the discriminating compounds are and there is no discussion of the results with respect to published literature.

The conclusion reports a 99% accuracy by the FT-MIR spectroscopy coupled with chemometrics. However, to result and discussion informing about accuracy calculation. Additionally in the abstract, the authors claim for 100% accuracy.

Round 2

Reviewer 2 Report

na

Author Response

Reviewer 2 has no comments or suggestions.

Reviewer 3 Report

Line 174, the authors confused about using the validation term of sensitivity, selectivity, and specificity. Recognition is not mostly related to sensitivity but selectivity.
Response: This section was rewritten because reviewer 2 requested it.

Please note that it is not a request by the reviewer to correct the information provided in the ms. The author has the liberty to take or not the given suggestions. Additionally, in this revised section, why the author prefers the term specificity instead of selectivity? Please check again the terms for the parameters of method validation to get a better understanding between selectivity and specificity.

Author Response

RESPONSE: The word specificity was changed for selectivity (lines 123, 196 and 202).
